**Knowledgebase and Database Resources**

# PomBase in 2026: expanding knowledge, modeling connections

Pascal Carme (ID) ,[1,2,*] Kim Rutherford (ID) ,[1] Jürg Bähler (ID) ,[2] Juan Mata (ID) ,[1] Valerie Wood (ID) [1,*]

[1]Department of Biochemistry, University of Cambridge, Cambridge CB2 1GA, United Kingdom
[2]Institute of Healthy Ageing, Department of Genetics, Evolution and Environment, University College London, London WC1E 7JE, United Kingdom

*Corresponding authors: Pascal Carme, Department of Biochemistry, University of Cambridge, Cambridge CB2 1GA, United Kingdom. Email: pascal@pombase.org; Valerie Wood, Department of Biochemistry, University of Cambridge, Cambridge CB2 1GA, United Kingdom. Email: val@pombase.org

PomBase is the model organism database dedicated to the fission yeast *Schizosaccharomyces pombe*. In this update, we outline recent progress in literature curation, the introduction of new tools, and enhancements designed to better support the research community. We highlight our recent effort to curate biological pathways and modules as causal networks using Gene Ontology—Causal Activity Modelling and describe new features that utilize these models to guide and inform hypothesis-driven research.

Keywords: model organism; model organism database; MOD; fission yeast; *Schizosaccharomyces pombe*; database; knowledgebase; biocuration; causal activity modelling; GO-CAM

## Introduction

As a major unicellular eukaryotic model, *Schizosaccharomyces pombe* has been central to our understanding of the fundamental processes governing the life cycle of all eukaryotic cells. For instance, many of the key factors that control the highly conserved eukaryotic cell cycle were initially identified in *S. pombe* (Nurse 2020). Fission yeast has also been instrumental in studying other fundamental aspects of cell biology, such as cell division (Mangione and Gould 2019), heterochromatin assembly and epigenetic inheritance (Grewal 2023), and pre-mRNA splicing (Fair and Pleiss 2017), among many others. Recent studies in *S. pombe* have identified and further characterized mechanisms and factors that are conserved from yeast to mammals, with notable examples including aging (Anver et al. 2024), autophagy (Zou et al. 2023), and epigenetic memory (Charlton et al. 2024; Toda et al. 2024; Yu et al. 2024). After a period of relative neglect, mitochondrial research is experiencing a revival, with recent studies uncovering diverse processes tied to mitochondrial diseases such as mitochondrial gene expression (Dinh and Bonnefoy 2024; Luo et al. 2024), oxidative phosphorylation (Malecki et al. 2020; Moe et al. 2023), and novel insights into mitochondrial cristae architecture (Kumar et al. 2024). Finally, *S. pombe* is also used to study the impact of disease-associated mutations at the molecular level (Qin et al. 2024; Lu et al. 2025), highlighting its potential as a genetic model for translational research on human disease-associated genes (Zhang et al. 2023).

PomBase (https://www.pombase.org) is the Model Organism Database (MOD) dedicated to the curation and integration of experimental data from studies performed in *S. pombe*. Through manual curation of biological literature, as well as standardization and organization of functional annotations, PomBase provides an up-to-date and accessible knowledgebase that connects genotypes to phenotypes, and gene products to functional information.

As the volume and complexity of biological data continue to grow, the need for reliable, expert-curated resources has never been greater. In this report, we highlight PomBase's recent progress in literature curation and the identification of unknown proteins. We also describe new and updated tools provided to our users. These improvements reinforce PomBase's role as an essential, expert-curated resource for researchers worldwide.

PomBase has recently prioritized representing biological processes as pathway models using the Gene Ontology—Causal Activity Modelling (GO-CAM) framework (Thomas et al. 2019). Pathway curation transforms our existing individual functional annotations into biological network representations, capturing cellular processes at the systems level. We report our progress in curating and implementing these pathway models in PomBase.

Finally, we outline our future directions, with a focus on developing large language models (LLM) training datasets and Artificial Intelligence (AI)-driven tools to enhance curation workflows.

## Curation
### Literature curation

Manual curation of experimental data is the primary source of knowledge for most MODs. Through this process, curators translate the results described in research publications into provenanced, gene-specific annotations that associate unique entities —such as genes and proteins—with functional information such as enzymatic activities, biological processes, cellular localization, or mutant phenotypes.

Biomedical ontologies are hierarchically structured vocabularies that define "terms" within a domain and the relationships between them. For example, the molecular function term "ATP-dependent protein folding chaperone" from the Gene Ontology (GO)

(Ashburner et al. 2000; The Gene Ontology Consortium 2023) is defined as the function of a protein which binds to another protein or a protein complex to assist in its folding, using the energy derived from ATP hydrolysis. Therefore, in the ontology, this term is a more specific instance—a "child"—of both the "ATP-dependent activity" and "protein folding chaperone" terms.

In ontology-based curation, entities are annotated with terms defined in the ontologies. This ensures consistency and interoperability across datasets and between databases. Additionally, by using precise, machine-readable terminology, ontology-based curation supports automated knowledge propagation between species (for example, Feuermann et al. 2025), and the development of computational tools for data analysis.

PomBase supports the curation of several annotation types, for both low-throughput hypothesis-driven and high-throughput genome-wide experiments, utilizing domain-specific ontologies. For example, molecular function, biological process, and cellular component annotations use the GO framework (Ashburner et al. 2000; The Gene Ontology Consortium 2023), phenotypes use the Fission Yeast Phenotype Ontology (FYPO) (Harris et al. 2013), and protein modifications use the Protein Ontology (Natale et al. 2017). Diseases associated with human orthologs of fission yeast genes are annotated using the Mondo disease ontology (Vasilevsky et al. 2025) and the PomBase *S. pombe*-human ortholog inventory (1,599 of the 3,637 *S. pombe* proteins with human orthologs are currently assigned to at least 1 disease term). Other curated data types include gene expression, genetic and physical interactions, orthology, and paralogy (Fig. 1). Phenotype annotations comprise the largest proportion of curated data in PomBase, representing over 40% of all annotations. Although 56% of phenotype annotations are derived from the integration of large-scale phenotype datasets, manually curated data from hypothesis-driven publications amount to over 85,000 annotations.

## The evolving nature of publication contents

All articles mentioning *S. pombe* or fission yeast in their abstract or keywords are retrieved from PubMed and loaded into the Canto curation and literature management tool (Rutherford et al. 2014) for processing (14,780 publications in Canto on 2025 December 16). All articles discussing experimental data obtained in *S. pombe* are in scope for hosting in the PomBase literature collection; however, only articles containing gene-specific data are curated.

A triage process separates the articles that are suitable for manual curation (7,035) from the articles that do not contain gene-specific experimental data. Articles that are not suitable for manual curation (7,745) are then classified into different categories. Among these, nearly half (3,258) are out of scope: they mention "*S. pombe*" or "fission yeast" but describe experiments in other species. Other categories include "Review or comment" (1,389 publications), "Method or reagent" (1,048), "Cell composition or WT feature" (378) for articles that present wild-type cellular process data with no gene-specific data, "Mutagenicity or toxicity study" (117), and "Not physically mapped" (111) for pre-genome sequencing studies of genes that were never mapped to a specific locus. Some of the "uncurated" articles are identified as providing data suitable for hosting in the PomBase instance of the JBrowse genome browser (32 publications not hosted on PomBase yet). The literature management tool is used to track the progress of articles through the community curation and review process.

After a period of relative stability with consistently high numbers of papers from 2005 to 2013, the number of *S. pombe* articles classified as curatable published each year has shown a slight decline over the past decade (Fig. 2a). However, this decrease in publication number is counterbalanced by, and perhaps related to, a large increase in the quantity of data reported in each publication. Small-scale hypothesis-driven articles are becoming increasingly knowledge-rich, with significantly increased numbers of genes being studied in each publication (a 3-fold increase since 2000) (Fig. 2b). This increase is largely due to the maturation of fields, as researchers extend their research programs to larger groups of genes and their context within broader biological pathways. The increase in the number of genes studied per publication has led to a corresponding increase in the number of annotations per publication (2-fold since 2000) (Fig. 2c). Additionally, as the *S. pombe* research community develops new high-throughput experimental tools, large-scale studies have also increased in size in recent years (Fig. 2d). These publications are curated as bulk annotation datasets imported into the database. Although the data from these studies are not reviewed individually, they still require formatting and standardization (for example, mapping the gene names to the corresponding systematic identifiers, using the relevant ontology terms and evidence types, or adding metadata to genome browser datasets), increasing the curators' workload. Overall, these advances in the way that knowledge is generated and presented have substantially increased the curatorial workload for both large- and small-scale studies despite the slight decrease in *S. pombe* publication numbers.

## microPublication: a simple way to share single experiments

In contrast to the growing complexity of traditional "long-form" publications, the *microPublication Biology* journal (https://www.micropublication.org/) accommodates data that might not fit the narrative of a longer, more focused article. They publish peer-reviewed, PubMed-indexed, 1-figure articles to rapidly disseminate valuable experimental results that would otherwise remain unpublished. These can include incidental findings (Basu et al. 2025), student projects (Hanna et al. 2025), negative results (Lopez Maury et al. 2024), or reproducibility of previously published data. PomBase collaborates with *microPublication* by participating in the editorial process and promoting the initiative to the *S. pombe* community. The number of *S. pombe* micropublications has increased steadily since the publication of the first one in 2021, reaching 53 publications in December 2025.

## Community curation

MODs are facing a major challenge: the increasing amount and complexity of data in hypothesis-driven publications make it difficult to curate knowledge at the same rate it is being produced. For a small team like PomBase (2 curators, 1 developer), keeping pace with new publications and managing the backlog of older curatable publications, along with other curation-related tasks, is especially demanding. To make this task more manageable, PomBase relies heavily on the research community to curate their newly published work.

Publication authors can directly curate their own data using the Canto online curation platform, an intuitive and user-friendly interface developed by PomBase (Rutherford et al. 2014). Community-curated articles are then reviewed by a PomBase curator to ensure that all annotations are consistent with our standardized curation principles and that curation is complete. Aside from freeing up significant time for PomBase curators to focus on other tasks, relying on the relevant experts to capture the data they published allows for productive author-curator dialogue, resulting in more accurate and comprehensive "gold-standard" annotations.

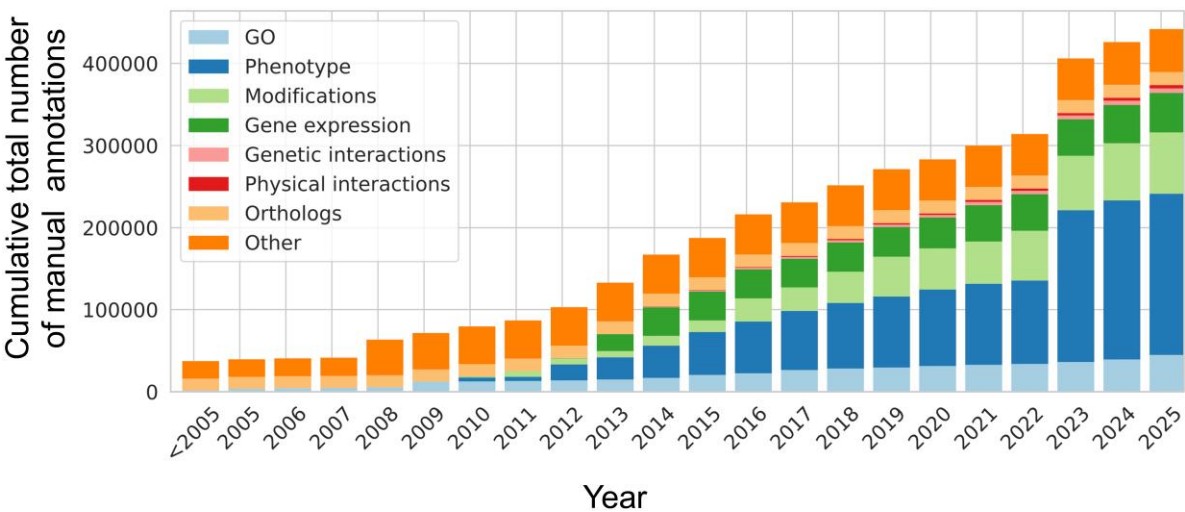

**Fig. 1.** Evolution of the numbers and types of annotations reported in PomBase over the years. These numbers include annotations manually curated by PomBase curators and the fission yeast community, but exclude annotations exported from other resources and annotations inferred from computational methods.

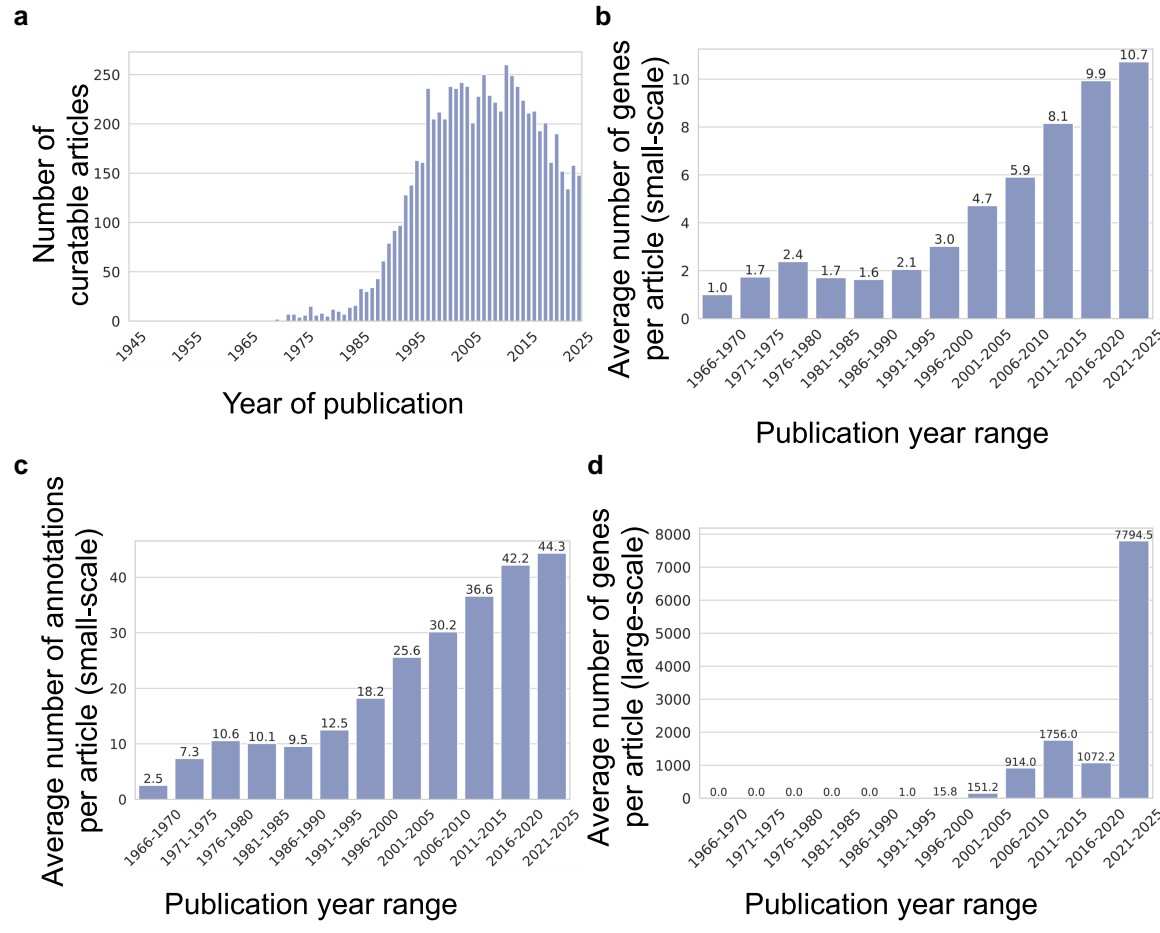

**Fig. 2.** Recent evolution in publication content. a) Number of curatable *S. pombe* articles published per year. b) Average number of genes annotated in small-scale experiments publications, binned by 5-year ranges of publication date. c) Average number of annotations in small-scale experiments publications, binned by 5-year ranges of publication date. d) Average number of annotations in large-scale experiments publications, binned by 5-year ranges of publication date.

Community curation also benefits authors. Their contributions can be cited in data-sharing activities and data-management plans required by funders, and they gain a better understanding of how to use data in PomBase (Lock et al. 2020). In addition, community-curated articles may be featured in the PomBase "Research Spotlights" section, increasing

**Table 1.** Annual numbers of papers curated in Canto by PomBase staff and by community contributors, along with the total annotations generated each year, from 2012 to 2025 December 16. Any paper that is fully or partially curated by a community member is considered a community-curated paper.

| Year | Publications curated | | | Annotations | |
|---|---|---|---|---|---|
| | All | PomBase | Community | PomBase | Community |
| 2012 | 367 | 356 | 11 | 3,558 | 121 |
| 2013 | 455 | 376 | 79 | 6,045 | 647 |
| 2014 | 902 | 822 | 80 | 10,895 | 891 |
| 2015 | 572 | 491 | 81 | 11,281 | 1,864 |
| 2016 | 426 | 289 | 137 | 10,089 | 2,275 |
| 2017 | 287 | 196 | 91 | 7,054 | 1,312 |
| 2018 | 309 | 180 | 129 | 5,847 | 3,938 |
| 2019 | 343 | 217 | 126 | 5,793 | 1,845 |
| 2020 | 228 | 107 | 121 | 4,597 | 2,189 |
| 2021 | 201 | 86 | 115 | 3,514 | 2,169 |
| 2022 | 132 | 67 | 65 | 1,246 | 3,009 |
| 2023 | 205 | 125 | 80 | 2,672 | 2,534 |
| 2024 | 281 | 171 | 110 | 7,410 | 2,305 |
| 2025 | 97 | 29 | 68 | 2,336 | 1,882 |
| Total: | 4,805 | 3,512 | 1,293 | 82,337 | 26,981 |

the visibility of their work, likely increasing the number of citations.

Since 2012, authors have contributed ~25% of the 109,318 new annotations captured in Canto from low-throughput hypothesis-driven experiments (Table 1). With the decrease in PomBase curation—a direct consequence of decreased funding over the past decade—the number of community-curated publications has been rivaling that of staff-curated articles in recent years. In total, more than a quarter of the curated publications displayed on PomBase underwent community curation (1,293 out of 4,805; 26.9%; Table 1 and Fig. 3a). Most importantly, the proportion of authors being assigned a community curation session and accepting the assignment has steadily increased since the implementation of Canto, recently reaching 57.2% (2025 December 16) (Fig. 3b).

## Curation progress

Since November 2023, the number of curated articles has increased from 4,440 to 4,804. During the same period, the total number of curatable publications has increased from 6,724 to 7,035. Thus, the overall percentage of curated papers has slightly increased since our previous database update (68.3%, compared with 66% previously) (Rutherford et al. 2024). New publications are largely curated by the community, allowing PomBase curators to focus on the backlog of uncurated publications from before the inception of PomBase in 2012 (Fig. 3c; 73.9% of pre-2012 articles curated as of December 2025).

## GO-CAM: from data to knowledge
### Introducing GO-CAM in PomBase

PomBase and its predecessors have used the GO resource to describe the molecular functions, biological processes, and cellular locations of gene products for over 20 years (The Gene Ontology Consortium 2023). GO-CAM provides a system built on GO to represent biological processes as a causal network by connecting GO Molecular Function terms (activities) encoded by individual gene products (Thomas et al. 2019). In a GO-CAM pathway, activities are also connected to GO Biological Process terms (representing the pathway they are part of) and GO Cellular Component terms (the localization in which they occur). This allows curators to create models representing the causal interactions between the molecular activities enabled by the different gene products acting in a biological process (Fig. 4a). Since they are based on GO annotations, GO-CAM pathways are fully provenanced, with every association and connection being supported by experimental or inferred evidence.

Given the usefulness of understanding gene function within the context of biological processes, PomBase has made a strong commitment to curating biological pathways through GO-CAM, with an ambitious goal of providing a comprehensive causal model of a eukaryotic cell within the next 3 years. GO-CAM pathways will support numerous computational applications exploiting the causal relationships between entities. Potential use cases include mapping lists of genes onto pathways, predicting the effects of function perturbations, querying through logical reasoning (e.g. "all processes causally downstream of a specific activity during mitotic interphase"), inferring missing relationships, comparative studies either between or within species, and supporting conversion into mathematical models to simulate pathway dynamics.

All PomBase-curated GO-CAM pathways are displayed on the relevant GO Biological Process term pages (e.g. adenylate cyclase-activating glucose-activated G protein-coupled receptor signaling pathway https://www.pombase.org/term/GO:0010619), as well as on the pages of all the genes enabling an activity in the pathway (e.g. *cyr1* https://www.pombase.org/gene/SPBC19C7.03). We have implemented a pathway viewer using Cytoscape (Shannon et al. 2003) to display the GO-CAM pathways on PomBase (Fig. 4b). This viewer allows intuitive manipulation of the nodes and of the field of view by clicking and dragging, zooming in or out by using the "+" and "−" buttons on the right side or the mouse scroll, and exporting the pathway as a vector graphics image by clicking on the "View as SVG" button.

In January 2025, we publicly displayed the first set of 36 PomBase GO-CAM pathways. We have since implemented a total of 115 biological pathways (as of December 2025), spanning 1,296 unique gene products. This represents a coverage of more than 25% of the proteome of *S. pombe*, and 29% of the proteins with a known process. The complete list of PomBase-curated GO-CAM pathways is available at https://www.pombase.org/gocam/model-list.

Since metabolic pathways are generally well characterized and well conserved, we prioritized their curation and were able to model several important pathways over a relatively short period. We also prioritized modeling mitochondrial processes, as part of a collaboration with the *Drosophila* MOD FlyBase to produce a complete interconnected model of a self-contained biological system using GO-CAM. So far, we have modeled the activities of 483 mitochondrial proteins, covering about 67% of the 722 protein-coding genes annotated to the GO Cellular Component term "mitochondrion" on PomBase (74 of the remaining 239 have no known function).

## Using pathway models to support and guide future research

In addition to pathway curation, we have developed several GO-CAM-related datasets, views, and tools that can be accessed from our GO-CAM pathways page (https://www.pombase.org/gocam).

We provide a list of missing activities or "pathway holes": activities that have been identified or should exist in *S. pombe* but are not currently annotated to any gene product. For example, in the nitrogen cycle metabolism, the allantoinase activity was

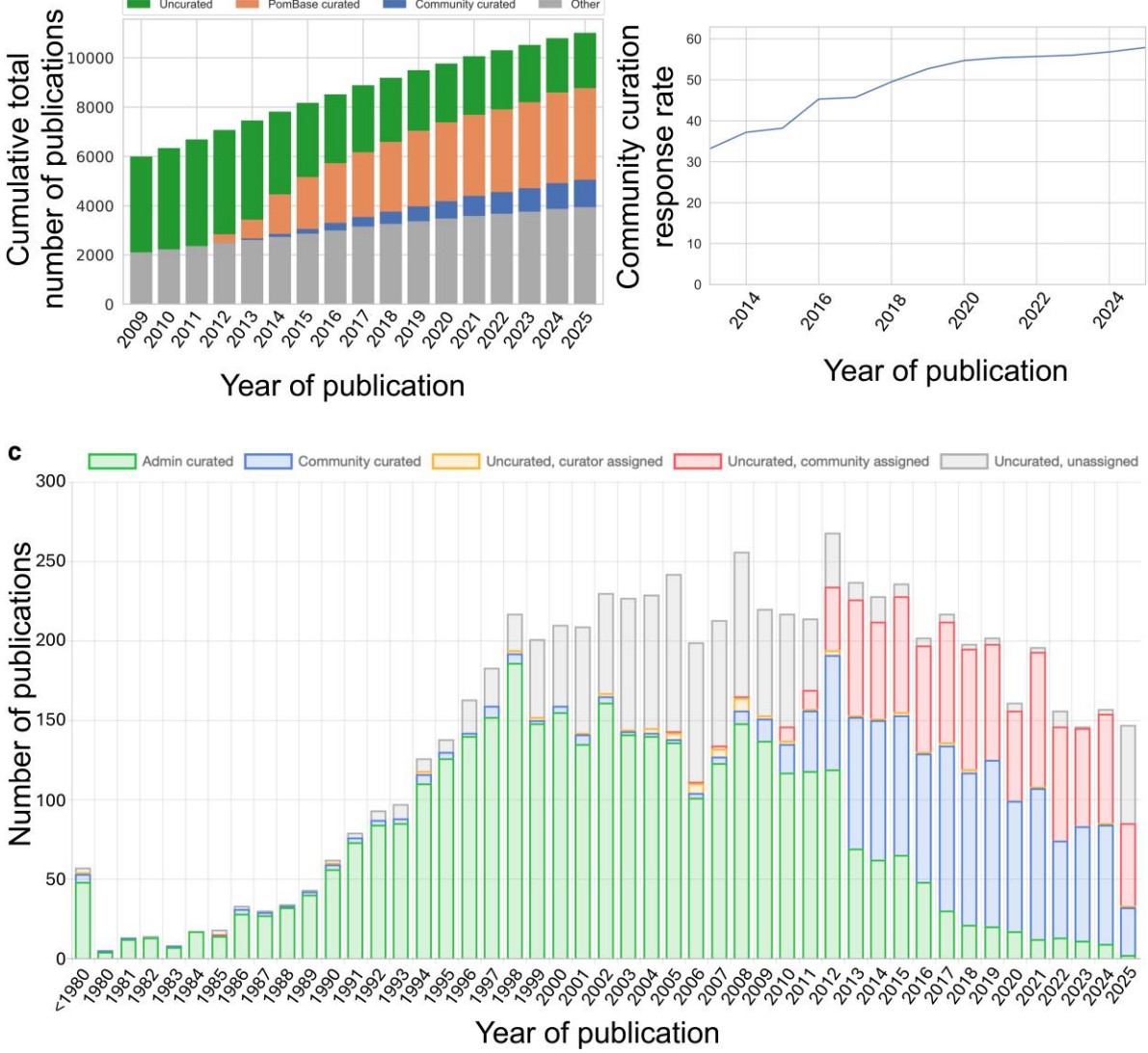

**Fig. 3.** Community curation metrics. a) Cumulative publication status of the in-scope PomBase literature. The "Other" category represents publications that are in scope but contain no gene-specific data (see section "The evolving nature of publication contents" for more details). b) Evolution of the percentage of community curation requests accepted by authors, depending on the year of publication, since 2013. c) Curation status of articles depending on the year of publication. Five categories are shown: (1) admin curated (green): papers curated exclusively by PomBase staff, who concentrate on pre-2012 "legacy" publications; (2) community curated (blue): curated papers with annotations contributed by authors, mostly papers published after the release of Canto in 2012 The community provides first-pass curation for most new articles before reviewing by PomBase curators; (3) uncurated, curator assigned (yellow): publications assigned to a PomBase curator but not yet curated; (4) uncurated, community assigned (red): publications assigned to a community member but not yet curated; and (5) uncurated, unassigned (gray): publications not yet assigned for curation.

previously experimentally detected in *S. pombe* through a classical genetic screen and attributed to a gene named *all1* (Kinghorn and Fluri 1984). However, *all1* was never physically mapped to a genomic locus, and no obvious allantoinase ortholog has been identified in *S. pombe*. Consequently, despite the experimental detection of an allantoinase activity in *S. pombe* more than 40 years ago, no gene has yet been associated with this function. One candidate protein is the potentially horizontally transferred hydrolase Hhy1, an uncharacterized member of the Amidohydrolase-related PF01979 family, to which many other allantoinases belong (Holm and Sander 1997). The list of missing activities also reports enzymatic activities for which both the substrate and product are detected, but the activity could not be attributed to a specific protein. These include the reciprocal glycerol kinase and glycerol 3-phosphatase activities, the 6-pyruvoyltetrahydropterin synthase activity, and the N-acylsphingosine amidohydrolase activity. Most of the missing

activities are unidentified transmembrane transporters specific for chemicals that are created and consumed in different locations. To encourage hypothesis-driven research on unknown proteins, the missing activities are compiled into a resource detailing, for each activity, its substrate, product, localization, and the pathway it is required in. The list is available at https://www.pombase.org/gocam/missing-activities.

We recently introduced 2 new interactive views designed to support the exploration and use of GO-CAM pathways: the "Mega Model" and "Summary Map." The "Mega Model" combines all currently available pathways into a single interconnected model (Fig. 5a). The "Summary Map" provides a high-level visual overview of the causal connections between individual GO-CAM pathways, without displaying their internal structure (Fig. 5b).

PomBase users can also analyze gene lists (imported or query-generated) using GO-CAM pathways, via 2 different options. The

**a**

### Standard GO annotation: associates a gene product to a GO term

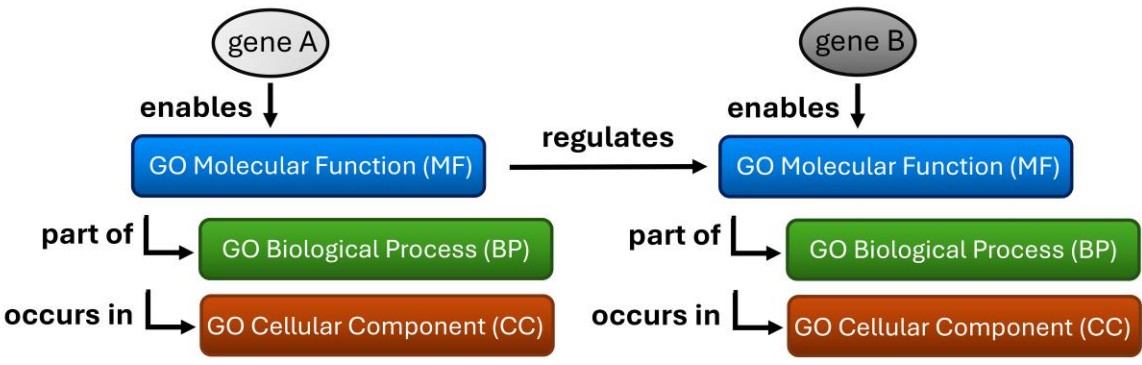

### GO-Causal Activity Modelling: network of standard annotations

**b**

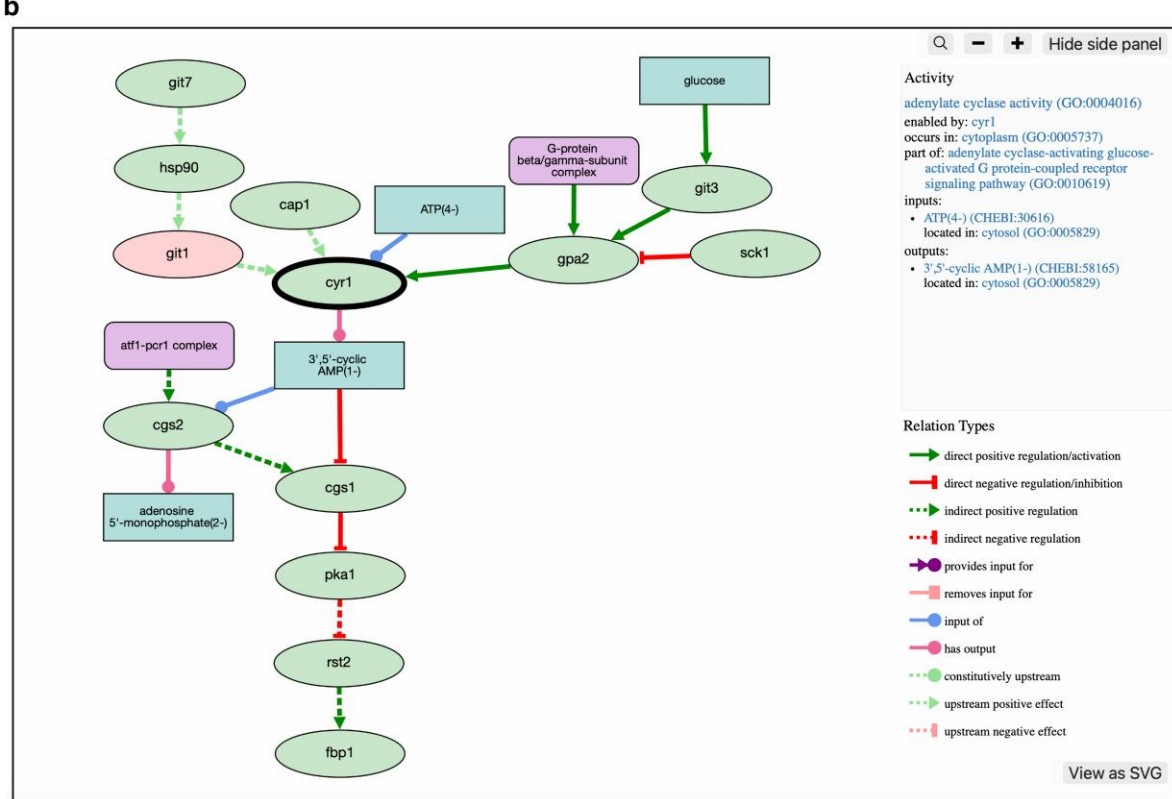

**Fig. 4.** GO-CAM pathways and their implementation in PomBase. a) Diagram representing the difference between standard GO annotations (top) and the GO-CAM framework (bottom). b) Representative image of a GO-CAM pathway, here the "adenylate cyclase-activating G protein-coupled receptor signaling" pathway, as displayed on PomBase. The left side shows an overview of the causal network representing the pathway. Selecting an entity (here, the cyr1 gene product) displays all the annotations associated with that entity in the model in the top panel on the right side of the viewer.

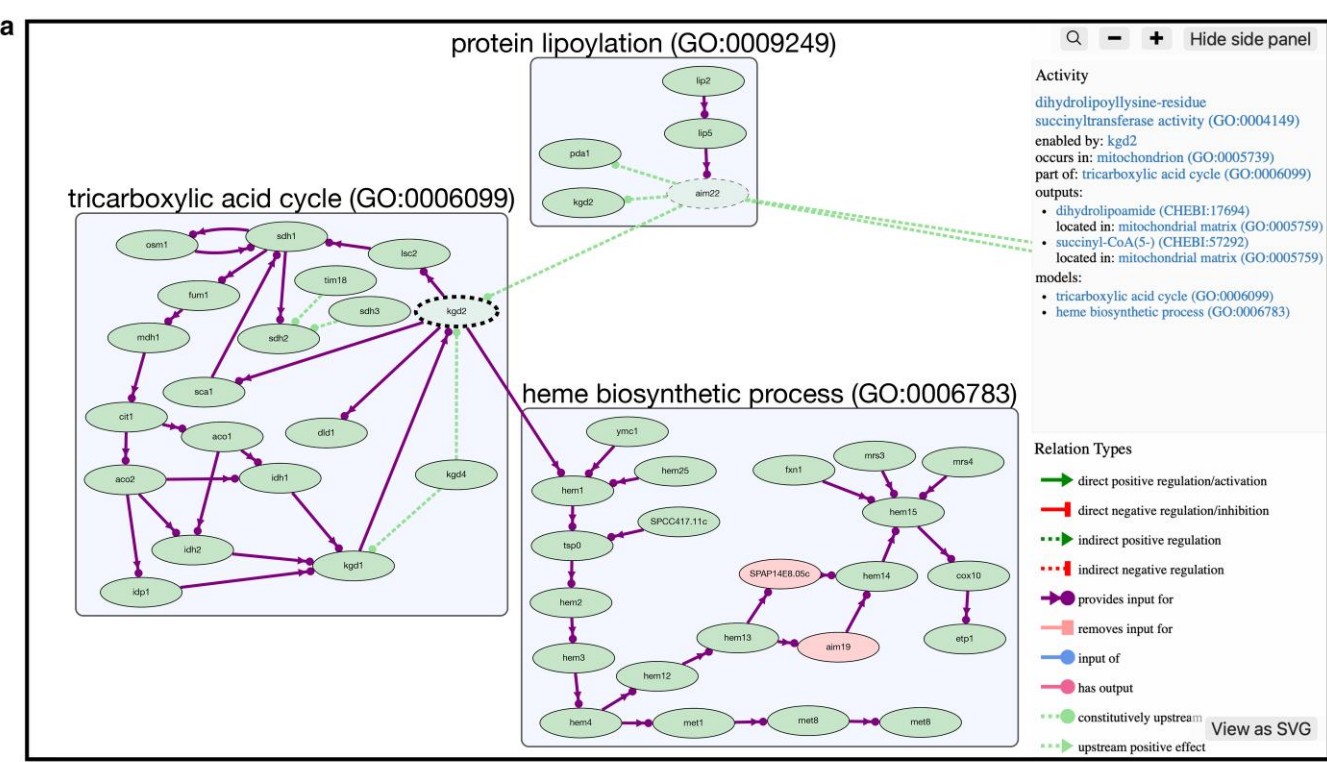

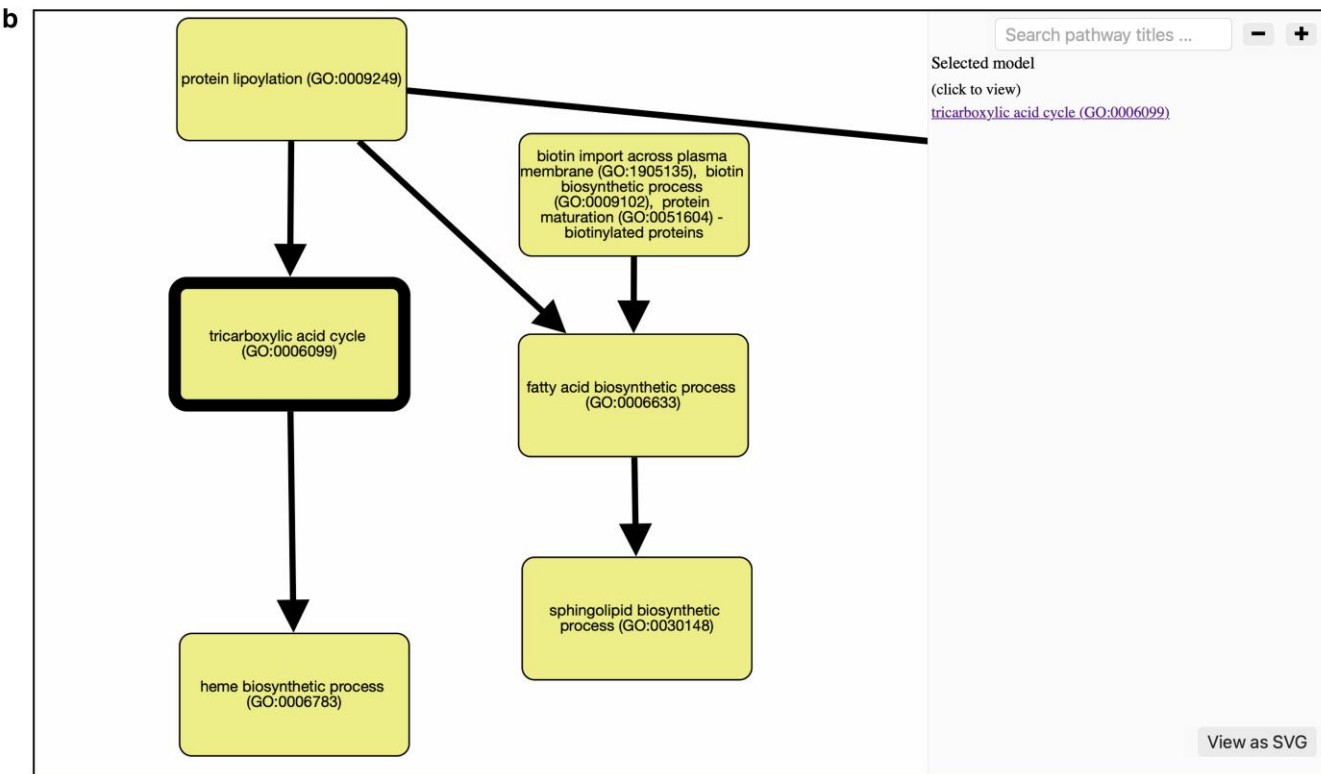

**Fig. 5.** New viewers to fully exploit the potential of GO-CAM. a) Partial view of the Mega Model, as displayed on PomBase. This viewer shares the same functionalities as the pathway viewer shown in Fig. 4b. b) Partial view of the Summary Map, as displayed on PomBase. Selecting a process displays a link to the corresponding pathway in the right-hand panel.

"View pathway coverage" option allows users to see which curated pathways contain genes from their list and highlight these genes in the pathway model (Fig. 6). The "View coverage in Mega Model" option highlights all the genes from the list in a truncated version of the Mega Model displaying only the pathways that contain genes from the list. These new tools should enable our community to easily assess the biological context in which their genes of interest function and formulate new hypotheses and experiments.

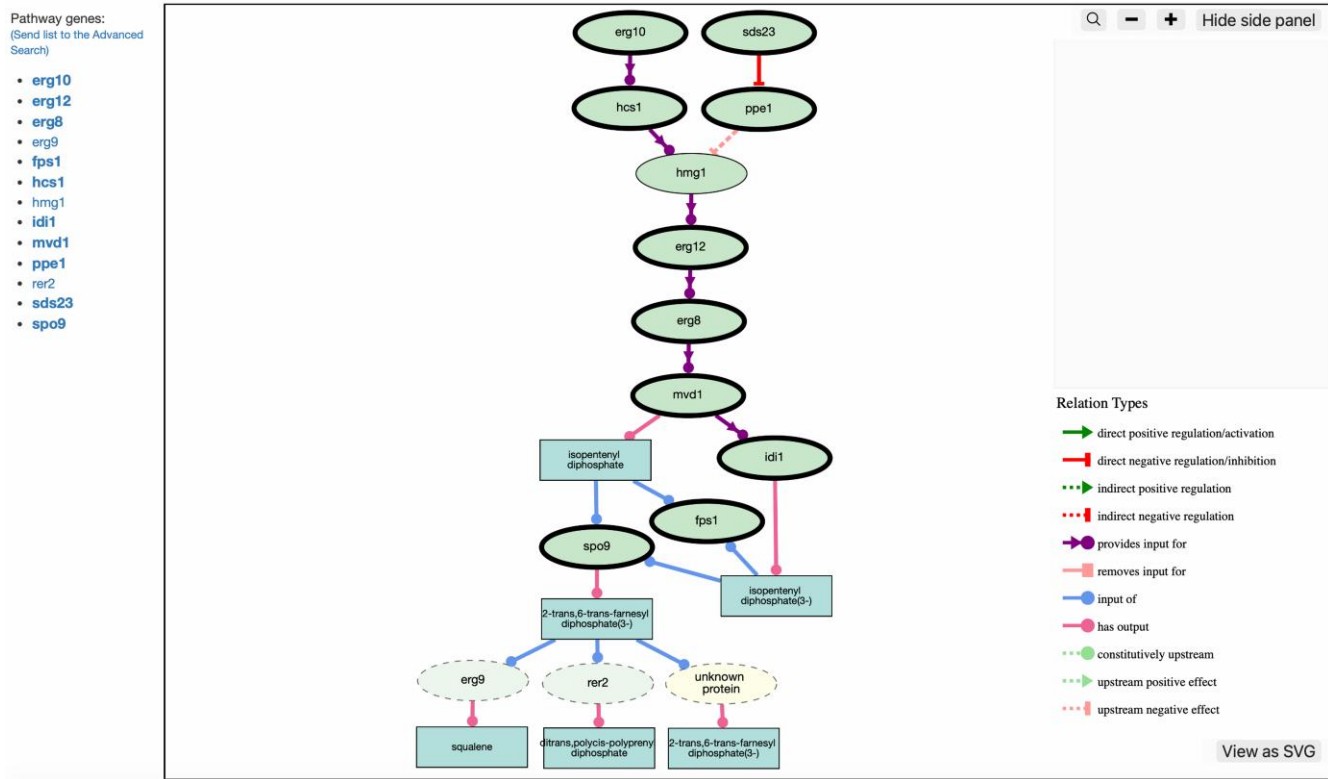

**Fig. 6.** Partial view of the "isoprenoid biosynthetic process" pathway with the gene products associated with the GO cellular component term "cytosol" highlighted (in both the graphical view and the gene list on the left) using the "View pathway coverage" option.

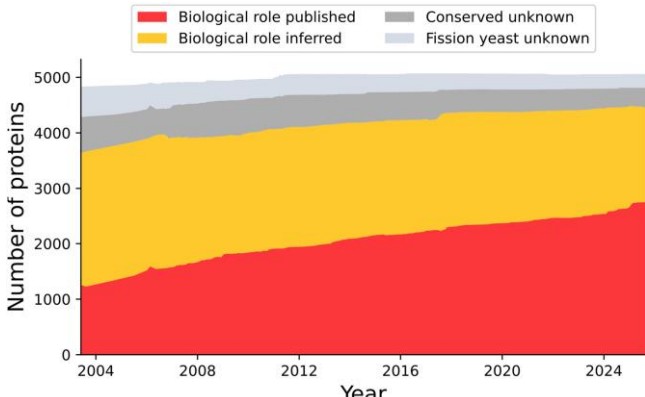

**Fig. 7.** Evolution of the characterization status of *S. pombe* protein-coding genes since 2004. We separate genes with a known role into (1) "Biological role published" (red), where the process is characterized in hypothesis-driven experiments in *S. pombe*; and (2) "Biological role inferred" (yellow), where the process is inferred from experimental results obtained in other species. We separate "unknown" proteins into (1) "Conserved unknown" (dark gray), for genes present in species outside the Schizosaccharomyces clade; and (2) "Fission yeast unknown" (light gray), for genes restricted to the fission yeast clade.

## Updated features and new datasets
### Into the unknown

The characterization of proteins of unknown function, the "unknome," has become an area of growing interest in recent years. The cellular function of a protein is often complex and multifactorial, inferred from the integration of diverse data types such as phenotypes, genetic interactions, and localization, derived from various orthogonal sources. Consequently, on an individual scale, determining if the function of a protein is known or unknown can easily turn into a qualitative assessment of how well characterized this function is. However, such an approach is not practical for an objective, genome-wide inventory of the number of genes of unknown functions in an organism. To address this challenge, PomBase defines an "unknown" gene as one which has no annotation to any broad GO Biological Process term (referred to as a GO "slim") describing its cellular role (the rationale for this decision is described in Wood et al. 2019 and Rutherford et al. 2024). This approach provides us with a simple, yet biologically meaningful binary metric to determine the number of *S. pombe* genes with an unknown role. As of December 2025, the number of *S. pombe* protein-coding genes of unknown role stands at 616 out of 5,058 genes. Therefore, around 87% of the proteome (4,395 genes) is associated with a Biological Process "slim" term, of which over 60% are based on experimental data from publications on *S. pombe* (2,732 genes with a published biological role) (Fig. 7).

The experimentally derived GO data associated with published genes are used to support functional inferences for orthologous genes in other species, making them useful to the broader research community (Feuermann et al. 2025). Similarly, the biological functions of 1,687 *S. pombe* proteins are inferred from experimental data associated with their orthologs and curated by other knowledgebases.

We provide a dashboard to access the *S. pombe* unknown gene sets (https://www.pombase.org/status/protein-status-tracker). Out of the 616 "unknown" *S. pombe* protein-coding genes, only 110 are specific to *S. pombe,* 369 are conserved outside of the *Schizosaccharomyces* genus, and 123 "priority unstudied genes" are conserved in humans and other mammals.

## Improved genome browser and new datasets

We have recently implemented the updated version of the JBrowse genome browser, JBrowse 2 (Diesh et al. 2023), for use with PomBase-curated datasets (https://www.pombase.org/jbrowse2). JBrowse 2 introduces multiple new features, such as the ability to display several genomic sequences (eg regions of different chromosomes) in a single view, toggles for displaying the forward, reverse, and translated sequences on the reference sequence dataset, and support for new data types, including Hi-C datasets.

We have added 6 new genome-wide datasets (337 new tracks) to JBrowse since our previous update, including upstream open reading frames coordinates (Duncan and Mata 2017; Moro et al. 2021) and alternative transcripts (Montañés et al. 2022). We have engaged with the authors of a recently published inventory of transcription factors binding sites (Skribbe et al. 2025) to add chromatin binding datasets (323 tracks) to the genome browser, as well as new DNA-binding consensus sequences for 38 transcription factors to the "DNA binding site consensus sequences" page. We also provide external links to the TFexplorer website (https://data.fmi.ch/TFexplorer/) on the transcription factors gene pages and the publication page. With the addition of these new datasets, as well as tracks that map transcription factor binding sequences to the entire genome, which we are currently implementing, the total number of hosted tracks is expected to reach 717 in the coming months.

## Outreach

During the global lockdowns caused by the COVID-19 pandemic, the S. *pombe* research community started a series of online seminars called pombeTalks. These community-driven seminars have remained a regular gathering for S. *pombe* researchers around the world for the past 5 years, with meetings now held monthly (https://www.pombase.org/community/pombetalks). Taking advantage of this existing stage, we initiated a series of live demonstrations of specific features present on PomBase to help the community engage with some of the lesser-known tools we provide. We hosted two 10-min presentations named "Just one thing," showcasing our Advanced search and query builder functionalities (https://www.pombase.org/query) and our gene lists visualization options back in January and February 2025, and we plan to continue demonstrating more PomBase tools and features in future editions of pombeTalks.

We also continue to engage with our community via the pombelist email group (1,266 members), Slack, LinkedIn, BlueSky, Mastodon, the PomBase news feed, and presentations at the Fission Yeast International Meetings.

## Conclusions and future directions

We will continue to focus on the curation of biological pathways and modules and develop new features to fully exploit the potential of GO-CAM to support research. The high-quality, detailed annotation of gene functions jointly curated by PomBase and the S. *pombe* research community has enabled us to curate a significant number of pathways in just 15 months, covering more than 20% of the proteome.

Thus, S. *pombe* has the highest percentage of proteins represented in GO-CAM pathways. It is also the eukaryotic organism with the smallest proportion of proteins with no biological process annotation (ie fewest "unknowns") (Wood et al. 2019; Xue and Rhee

2023; see https://genomeannotation.rheelab.org/#!/overview for current numbers). Based on proteome size, process coverage, and our current pathway coverage and curation trajectory, we envision that S. *pombe* could provide the first complete causal model of a eukaryotic cell.

The emergence of AI, particularly LLMs, provides the biocuration community with opportunities to streamline curation tasks using AI-based methods. However, a common limitation of LLMs is the need for massive quantities of high-quality data for their training and benchmarking. The PomBase curator-reviewed community curation, supported by numerous quality control checks, provides a large dataset of accurate, consistent, and comprehensive GO and phenotype annotations for a substantial number of hypothesis-driven research publications (4,798). These gold-standard annotation datasets can serve as a reliable ground truth for training and benchmarking models designed to learn the nuanced criteria used by expert curators and partially automate the curation process. Over recent years, we have, when possible, recorded either text spans from the publications or figure numbers supporting the annotations (19,719 citation-associated annotations) during the curation or reviewing of community-curated articles. We have made these citation-associated annotations available as an AI benchmarking dataset, alongside other datasets useful for benchmarking, named entity recognition and literature classification models.

We are working closely with AI experts to integrate AI-generated outputs into our workflows for literature triage, gene summarization, named entity recognition, data extraction, and ontology development and quality review. Importantly, all AI-based data extraction (gene product to ontology term associations) will be reviewed by curators before incorporation into PomBase. These efforts aim to harness the strengths of AI-based models to streamline the curation process, while preserving the expert oversight that underpins the accuracy and quality of PomBase annotations.

By being at the forefront of innovation in curatorial practice, PomBase will continue to provide new ways to explore existing knowledge to support researchers in driving the discoveries of the future.

## Data availability

The PomBase website is updated daily, and monthly archived versions of all PomBase data files are available. We have recently updated our monthly releases architecture to provide a clearer layout and full documentation for each release. Each folder now contains a README file that details the folder's content, making it easier to retrieve specific files in the release. Along with this new site structure, we now support a permanent URL to the latest monthly release (https://www.pombase.org/latest_release/), and an archive of previous releases (back to 2018) in the new archive format (https://www.pombase.org/monthly_releases/). All PomBase code is available under an open-source license from the PomBase GitHub organization (https://github.com/pombase), where each major aspect of the project—including curation, website, Chado database, FYPO, and Canto—has a dedicated repository.

## Acknowledgments

PomBase thanks the fission yeast community for contributing datasets and literature curation and for providing valuable feedback on the website; our advisory board, Li-Lin Du, Kathleen Gould, Sabina Leonelli, Sophie Martin, Alison Pidoux, and Sigurd Braun for their

continual support; Andrew Green and Chris Mungall for advice on AI; Daniela Raciti, Karen Yook, and their team for providing the *S. pombe* microPublication platform and ongoing support; Helen Attrill, Steven Marygold, and Rossana Zaru for many discussions on GO-CAM modeling; Charlie Hoffman, Sarah Sabatinos, and Sarah Lambert for supporting our microPublication editorial process; Pascale Gaudet, Kimberly Van Auken, and the GO consortium for implementing our recommendations for improving GO annotations and ontology content; Sandra Orchard for adding *S. pombe* complexes to The Complex Portal; Snezka Oliferenko for overseeing the community submissions for JaponicusDB; Jon Hollis, Matthew Fairbairn, and team at the Babraham Institute for hosting and supporting the PomBase website.

## Funding

This research was funded in whole by the Wellcome Trust (grant number 218236/Z/19/Z to J.M.). For the purpose of open access, the author has applied a CC BY public copyright license to any author-accepted manuscript version arising from this submission.

## Conflicts of interest

None declared.

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

*Editor: T. Harris*