## [Peer Review File · Genetics]

PomBase in 2026: Expanding Knowledge, Modelling Connections

Pascal Carne, Kim Rutherford, Jürg Bähler, Juan Mata, and Valerie Wood

NOTE: The reviews and decision letters are unedited and appear as submitted by the reviewers.

In extremely rare instances and as determined by a Senior Editor or the EIC, portions of a review may be redacted. If a review is signed, the reviewer has agreed to no longer remain anonymous.

The review history appears in chronological order.

Review Timeline:

Submission Date:	2025-10-03
Editorial Decision:	2025-12-03
Revision Received:	2025-12-18
Accepted:	2025-12-23

December 3, 2025

RE: GENETICS-2025-308093

Dear Dr. Wood:

I am pleased to accept your manuscript titled "PomBase in 2026: Expanding Knowledge, Modelling Connections" for publication in GENETICS, pending minor revision.

Please submit your revision along with a brief description of how you modified the manuscript in response to the reviewers' concerns and suggestions (which can be viewed at the bottom of this email. In particular, multiple reviewers commented on the legibility of the figures. Hopefully, these can be approved during resubmission.

I expect you should be able to submit a revised manuscript within 30 days. A suitably revised manuscript will be acceptable for publication; I don't expect to send it out for review.

When revising the ms., please make an effort to shorten it, because that almost always improves a manuscript. We urge authors to heed the advice of Strunk and White: "omit needless words"¹. Follow this link to submit the revised manuscript: Link Not Available

Thank you for submitting this story to Genetics.

Sincerely,

Todd Harris
Associate Editor
GENETICS

Approved by:
Paul Sternberg
Senior Editor
GENETICS

Reviewer comments:

Reviewer #1 :

This manuscript describes PomBase, the model organism database for fission yeast, and reports recent advancements in literature curation, community engagement, curation using GO-CAM, and other items.

The authors discuss the increase in data complexity and quantity in publications. They may want to include a few sentences about the challenges of curating high-throughput studies and how they are addressing them.

The authors describe their effective community outreach efforts. The authors state that most of the backlog of pre-2012 papers are now curated, but then also say that the robust community curation activities allow PomBase curators to focus on curating the backlog of uncurated publications from before the inception of PomBase in 2012.

The authors describe new viewers for the GO-CAM models. The figures (all figures) seem to be low-resolution and should be improved.

It would be helpful to include a JBrowse URL.

The pombeTalks do not seem to be available on YouTube. Nor the 'Just one thing' presentations. These would be good additions to the authors' channels.

There are a lot of figures. Is each one necessary?

Overall the manuscript is informative and a good contribution to the research and biocuration communities. The authors have done an excellent job with PomBase, and are to be thanked for their efforts in community engagement and their development of data curation processes.

Reviewer #2 :

Review of "PomBase in 2026: Expanding knowledge, modelling connections" by Carme, Rutherford, Bähler, Mata, and Wood for Genetics

This short article describes the various features of an open-access website about the fission yeast *S. pombe*. The authors describe the many features available for each of the ~5000 genes, from DNA sequence, activity, and protein structure to interaction with other genes and their products and homologs in other species. They give a historical background, showing how the site has grown by curation of the many papers on *S. pombe*, and discuss current and anticipated future additions to the site. It is a clear description of an impressive amount of information available for researchers working with *S. pombe* or other species, including mammals. The analysis in Figure 5 (GO-CAM) appears to be a new feature for *S. pombe* and should be useful to many researchers regardless of species studied. The publication of this article in Genetics would enhance its visibility, important for those working with *S. pombe* or any other species.

Some points could be clarified, including the following.

p. 3. Explain "Gene Ontology" and define "AI". The latter is defined on p. 22, but it would be better here.

p. 3, at the bottom. An example of functional information about genes and their products would be useful, just as examples given on the next page are.

p. 7, near the middle. Insert "PomBase" to read "hosting in the PomBase genome browser."

p. 16, near the middle. The sentence beginning with "Other enzymatic ..." is too complex. One might insert "for which" to read "...detected, but for which no protein..." but splitting the sentence into two would probably be better. (This is one of the few examples in which the writing is not abundantly clear.)

p. 19, near the middle. Define or explain "slim".

Figures. Many of the labels are too small to read. Examples are the years (x axis) in Figs. 2A, 4, and 6. The texts in Figs. 4, 5, 7, and 8 are difficult or impossible to read. Black font would increase legibility. I think the figures should be legible when the published page is printed on ordinary letter-size paper.

Reviewer #3 :

This article provides a comprehensive update about the *S. pombe* database, pombase.org. This database provides an essential resource for the *S. pombe* community. The article is clearly written and there are mostly minor comments below.

1. Important! - The resolution of the figures is poor. Please make sure it is better for the actual publication.

2. page 2 - Please change the writing of the sentence about cell cycle to: "Many of the key factors that control the highly conserved eukaryotic cell cycle were initially identified in *S. pombe* (Nurse, 2020)."

3. page 2 - Charlton et al., cited as an example of epigenetic memory, was one of three *S. pombe* papers published at the same time on this topic, so all three should be cited here. The other two are Yu et al. (doi: 10.1016/j.cell.2024.07.006) and Toda et al. (doi: 10.1016/j.molcel.2024.07.002).

4. page 3, line 4 - insert "and" or "as well as" between literature and standardization.

5. Figure 2A, page 7 - It appears that the data only go up to 2020. Is anything more recent available? Also, from looking at the graph, I think it would be more accurate to say there was a consistent high number of publications from 2005-2013, which then declined.

6. Figure 2, page 7 - Please define low-throughput and high-throughput for how it is used here. Figure 2B and 2C and both called low-throughput in the legend, but only one has that label on the Y axis - please adjust.

7. page 16, line 5 - add a comma after "views"

We thank the reviewers for their comments and suggested improvement to our manuscript “PomBase in 2026: Expanding Knowledge, Modelling Connections”.

Regarding the general issue about the legibility of the figures, we sincerely apologise for the poor resolution of all the images in the previous version of the manuscript. We suspect that the loss of image quality happened during the upload onto GENETICS, as our version of the manuscript did not show the same issue. Regardless of the cause, we provide the text and figures as separate files for the revision which should prevent this issue from happening again, and we will do our best to ensure that this issue is not present in the final version of the manuscript before publication.

We have modified the manuscript to respond to the individual comments of each reviewer, and present our responses below.

Reviewer #1 : (changes highlighted in yellow in the revised text)

This manuscript describes PomBase, the model organism database for fission yeast, and reports recent advancements in literature curation, community engagement, curation using GO-CAM, and other items.

The authors discuss the increase in data complexity and quantity in publications. They may want to include a few sentences about the challenges of curating high-throughput studies and how they are addressing them.

- We have changed the end of the corresponding paragraph (page 7) to:

“Additionally, as the *S. pombe* research community develops new high-throughput experimental tools, large-scale studies have also increased in size in recent years (Figure 2D). These publications are curated as bulk annotation datasets imported into the database. Although the data from these studies are not reviewed individually, they still require formatting and standardisation (for example, mapping the gene names to the corresponding systematic identifiers, using the relevant ontology terms and evidence types, or adding metadata to genome browser datasets), increasing the curators' workload. Overall, these advances in the way that knowledge is generated and presented have substantially increased the curatorial workload for both large- and small-scale studies despite the slight decrease in publication numbers.”

The authors describe their effective community outreach efforts. The authors state that most of the backlog of pre-2012 papers are now curated, but then also say that the robust community curation activities allow PomBase curators to focus on curating the backlog of uncurated publications from before the inception of PomBase in 2012.

- We have removed the mention “Most of the backlog of pre-2012 papers are now curated” from the legend for Fig. 4, and changed the text in the following paragraph to further clarify this idea. The last sentence of this paragraph now reads:

“New publications are largely curated by the community, allowing PomBase curators to focus on the backlog of uncurated publications from before the inception of PomBase in 2012 (Figure 4C; 73.9% of pre-2012 articles curated as of December 2025).”

The authors describe new viewers for the GO-CAM models. The figures (all figures) seem to be low-resolution and should be improved.

- Please refer to our general response to the figure resolution issues above.

It would be helpful to include a JBrowse URL.

- A link to the following URL has been added to the text
<https://www.pombase.org/jbrowse2>

The pombeTalks do not seem to be available on YouTube. Nor the 'Just one thing' presentations. These would be good additions to the authors' channels.

- Recordings for pombeTalks season 6 episodes on YouTube are linked to from the pombeTalks page on PomBase (<https://www.pombase.org/community/pombetalks>), and recordings for previous episodes are available upon request to the pombeTalks organisers. Addition of the 'Just one thing' presentations as “video tutorials” to the PomBase website is planned for the near future.

The link to the PomBase pombeTalks page has been added to the article's text.

There are a lot of figures. Is each one necessary?

- The figures previously described as “*Figure 3: Evolution of the total number of S. pombe microPublications from 2021 to September 2025*” and “*Figure 6: PomBase GO-CAM curation metrics.*” have been removed as they were deemed not very informative.

Overall the manuscript is informative and a good contribution to the research and biocuration communities. The authors have done an excellent job with PomBase, and are to be thanked for their efforts in community engagement and their development of data curation processes.

Reviewer #2 : (Changes highlighted in green in the revised text)

Review of "PomBase in 2026: Expanding knowledge, modelling connections" by Carme, Rutherford, Bähler, Mata, and Wood for Genetics

This short article describes the various features of an open-access website about the fission yeast *S. pombe*. The authors describe the many features available for each of the ~5000 genes, from DNA sequence, activity, and protein structure to interaction with other genes and their products and homologs in other species. They give a historical background, showing how the site has grown by curation of the many papers on *S. pombe*, and discuss current and anticipated future additions to the site. It is a clear description of an impressive

amount of information available for researchers working with *S. pombe* or other species, including mammals. The analysis in Figure 5 (GO-CAM) appears to be a new feature for *S. pombe* and should be useful to many researchers regardless of species studied. The publication of this article in *Genetics* would enhance its visibility, important for those working with *S. pombe* or any other species.

Some points could be clarified, including the following.

p. 3. Explain "Gene Ontology" and define "AI". The latter is defined on p. 22, but it would be better here.

- We think that describing GO isn't necessary in the Databases and Resources section of *GENETICS*. We refer the readers to the recent GO updates, which describe GO in detail.

The final sentence of the Introduction, which introduces the idea of developing AI-assisted tools, has been changed to define the acronyms used:

"Finally, we outline our future directions, with a focus on developing Large Language Models (LLM) training datasets and Artificial Intelligence (AI)-driven tools to enhance curation workflows."

Additionally, we have improved the beginning of the "Introducing GO-Causal Activity Modelling in PomBase" section, rearranging the text in the first paragraphs of the section to clarify the relationship between GO annotations and GO-CAM.

p. 3, at the bottom. An example of functional information about genes and their products would be useful, just as examples given on the next page are.

- We have listed a few general examples of the types of functional information that can be associated with gene products at the end of this sentence.

p. 7, near the middle. Insert "PomBase" to read "hosting in the PomBase genome browser."

- Since the genome browser displayed on PomBase pages uses the open-source JBrowse platform, which is not an in-house product of PomBase, we have decided to further clarify this sentence, changing the it to:

"Some of the "uncurated" articles are identified as providing data suitable for hosting in the PomBase instance of the JBrowse genome browser"

p. 16, near the middle. The sentence beginning with "Other enzymatic ..." is too complex. One might insert "for which" to read "...detected, but for which no protein..." but splitting the sentence into two would probably be better. (This is one of the few examples in which the writing is not abundantly clear.)

- In line with the reviewer's suggestion to split this sentence into two simpler ones, we have changed this sentence to

"The list of missing activities also reports enzymatic activities for which both the substrate and product are detected, but the activity couldn't be attributed to a specific protein. These include the reciprocal glycerol kinase and glycerol

3-phosphatase activities, the 6-pyruvoyltetrahydropterin synthase activity, and the N-acylsphingosine amidohydrolase activity.”

p. 19, near the middle. Define or explain "slim".

- We have clarified what a GO 'slim' by changing the sentence to:

“To address this challenge, PomBase defines an ‘unknown’ gene as one which has no annotation to any broad GO Biological Process term (referred to as a GO ‘slim’) describing its cellular role”

Figures. Many of the labels are too small to read. Examples are the years (x axis) in Figs. 2A, 4, and 6. The texts in Figs. 4, 5, 7, and 8 are difficult or impossible to read. Black font would increase legibility. I think the figures should be legible when the published page is printed on ordinary letter-size paper.

- We have changed the size of the year labels in the mentioned figures, and Fig. 6 will be removed as mentioned in the response to Reviewer 1. The screenshots of the different GO-CAM pathway viewers have been replaced with zoomed-in views to make the texts inside more readable. We also expect that increasing the overall resolution of the figures will improve their legibility.

Reviewer #3 : (Changes highlighted in blue in the revised text)

This article provides a comprehensive update about the *S. pombe* database, pombase.org. This database provides an essential resource for the *S. pombe* community. The article is clearly written and there are mostly minor comments below.

1. Important! - The resolution of the figures is poor. Please make sure it is better for the actual publication.

- Please refer to our general response to this issue above.

2. page 2 - Please change the writing of the sentence about cell cycle to: "Many of the key factors that control the highly conserved eukaryotic cell cycle were initially identified in *S. pombe* (Nurse, 2020)."

- This sentence has been changed accordingly.

3. page 2 - Charlton et al., cited as an example of epigenetic memory, was one of three *S. pombe* papers published at the same time on this topic, so all three should be cited here. The other two are Yu et al. (doi: 10.1016/j.cell.2024.07.006) and Toda et al. (doi: 10.1016/j.molcel.2024.07.002).

- We thank the reviewer for pointing out this oversight. We have added the requested references to the article.

4. page 3, line 4 - insert "and" or "as well as" between literature and standardization.

-
- As suggested by the reviewer, the sentence has been changed to “Through manual curation of biological literature, as well as standardization and organization of functional annotations...”.

5. Figure 2A, page 7 – It appears that the data only go up to 2020. Is anything more recent available?

- The data presented in the mentioned plot went up to 2024 (2025 was left out because the year was incomplete at the time of writing). We have now replaced it with a version that includes the articles published in 2025.

Additionally, we have updated all the other graphs and numbers presented in the text and in Table 1 to their values at the time of the revision.

Also, from looking at the graph, I think it would be more accurate to say there was a consistent high number of publications from 2005–2013, which then declined.

- In line with the reviewer’s suggestion, we have changed this sentence to “After a period of relative stability with consistently high numbers of papers from 2005 to 2013, the number of *S. pombe* articles classified as curatable that are published each year has shown a slight decline over the past decade (Figure 2A).”

6. Figure 2, page 7 – Please define low-throughput and high-throughput for how it is used here. Figure 2B and 2C and both called low-throughput in the legend, but only one has that label on the Y axis – please adjust.

- We have replaced the ambiguous “low-throughput” and “high-throughput” terms in the text and Figure by the more self-explanatory terms “small-scale” and “large-scale”. We hope this clarifies what was intended here.

We have also added “(small-scale)” to the label on the Y axis for Fig. 2B, as suggested by the reviewer.

7. page 16, line 5 – add a comma after “views”

- The text has been modified accordingly.

December 23, 2025

RE: GENETICS-2025-308093R1

Dr. Valerie Wood
University of Cambridge
Biochemistry
Sanger Building
Old Addenbrooke's Site, Tennis Court Road
Cambridge, N/A CB2 1GA
United Kingdom

Dear Dr. Wood:

Congratulations, your manuscript titled "PomBase in 2026: Expanding Knowledge, Modelling Connections" is accepted for publication in GENETICS! Many thanks for submitting your research to the journal.

To Proceed to Publication:

1. Format your article according to GENETICS style: <https://academic.oup.com/genetics/pages/author-guidelines>
2. Ensure that you comply with data and community resource citation guidelines: <https://academic.oup.com/genetics/pages/author-guidelines#section-5-9-2>
3. Upload your final files at <https://genetics.msubmit.net>
4. Add oupsupport@scipris.com and genetics.oup@novatechset.com (or the domains @scipris.com and @novatechset.com) to your email program's "safe senders" list. You will be contacted by both at various points during the production process.

Notes:

- Your currently-accepted manuscript (unedited, as submitted, reviewed, and accepted) will be published at GENETICS and deposited into PubMed as an Advance Access article. Notify sourcefiles@thegsajournals.org before signing your license if you do not wish to publish your article via Advance Access.
- We invite you to submit an original color figure related to your paper for consideration as cover art. Please email your submission to the editorial office or upload it with your final files. You can submit a small-sized image for evaluation, and if selected, the final image must be a TIFF file 2513px wide by 3263px high (8.375 by 10.875 inches; resolution of 600ppi). Please avoid graphs and small type.
- After files are sent to Oxford University Press we use SciPris to manage article licensing and payment. If you do not have a SciPris account, you will receive an email from no-reply@scipris.com to sign up to use Oxford University Press' author portal. After logging in, follow the online instructions to sign your license and arrange any payment due.

If you have any questions or encounter any problems while uploading your accepted manuscript files, please email the editorial office at sourcefiles@thegsajournals.org.

Sincerely,

Todd Harris
Associate Editor
GENETICS

Approved by:
Paul Sternberg
Senior Editor
GENETICS

Review comments (if applicable):